# Evaluating the theoretical performance of aircraft wastewater monitoring as a tool for SARS-CoV-2 surveillance

Joseph W. Shingleton[ID]*, Chris J. Lilley, Matthew J. Wade[ID]

Analytics & Data Science Directorate, UK Health Security Agency, Nobel House, Smith Square, London, United Kingdom

* joseph.shingleton@ukhsa.gov.uk

**Data Availability Statement:** I have made all code and data associated with the paper available in the

## Abstract

Air travel plays an important role in the cross-border spread of infectious diseases. During the SARS-CoV-2 pandemic many countries introduced strict border testing protocols to monitor the incursion of the virus. However, high implementation costs and significant inconvenience to passengers have led public health authorities to consider alternative methods of disease surveillance at borders. Aircraft wastewater monitoring has been proposed as one such alternative. In this paper we assess the theoretical limits of aircraft wastewater monitoring and compare its performance to post-arrival border screening approaches. Using an infectious disease model, we simulate an unmitigated SARS-CoV-2 epidemic originating in a seed country and spreading to the United Kingdom (UK) through daily flights. We use a probabilistic approach to estimate the time of first detection in the UK in aircraft wastewater and respiratory swab screening. Across a broad range of model parameters, our analysis indicates that the median time between the first incursion and detection in wastewater would be approximately 17 days (IQR: 7–28 days), resulting in a median of 25 cumulative cases (IQR: 6–84 cases) in the UK at the point of detection. Comparisons to respiratory swab screening suggest that aircraft wastewater monitoring is as effective as random screening of 20% of passengers at the border, using a test with 95% sensitivity. For testing regimes with sensitivity of 85% or less, the required coverage to outperform wastewater monitoring increases to 30%. Analysis of other model parameters suggests that wastewater monitoring is most effective when used on long-haul flights where probability of defecation is above 30%, and when the target pathogen has high faecal shedding rates and reasonable detectability in wastewater. These results demonstrate the potential use cases of aircraft wastewater monitoring and its utility in a wider system of public health surveillance.

## 1. Introduction

Air travel has acted as a major conduit for international transmission of SARS-CoV-2 throughout the pandemic [1]. Almost all countries have, at some point, introduced degrees of border control, which have prevented inbound travel from specific locations, added requirements for evidence of a recent negative test or vaccination prior to departure, or required testing of

following public repository OSF: https://osf.io/sdy2f/.

**Funding:** The authors received no specific funding for this work.

**Competing interests:** The authors have declared that no competing interests exist.

passengers upon arrival [2–4]. Many of these requirements have been removed in a majority of countries due to a combination of factors, including successful vaccination programmes and a need to limit further economic disruption [4,5].

Prior to removal, border screening programmes provided rich data on the importation of SARS-CoV-2 [1], and hel6ped to map the international [6] prevalence of new variants [7]. With the reduction in large-scale screening of inbound passengers, many countries have begun to adopt monitoring of aircraft wastewater as a potentially non-intrusive alternative for monitoring pathogens entering a country or region [8–10].

There exist no standard protocols for aircraft wastewater surveillance and establishing the value of such a function for public health will require extended research and development. Subsequently, several significant limitations on its use remain, not least its acuity in capturing individuals harbouring pathogen(s) of interest, typically not an issue with gross population wastewater sampling. For wastewater monitoring to capture a single infected individual a series of conditions must be met. First, the individual must defecate during the flight—the probability of which varies with the length of the flight. Survey data suggests around 13% of individuals defecate during a short-haul (<6 hours) flight, increasing to 36% for long-haul flights [11]. Additionally, the individual must be shedding genetic markers (i.e., RNA or DNA) of the pathogen(s) in their faeces—the probability of which varies between and within pathogen types; however, it is thought to occur in between 30% to 60% of cases for SARS-CoV-2 [11,12]. Finally, the concentration of viral RNA in the aircraft wastewater must exceed the limit of detection (LoD) for Reverse Transcription Polymerase Chain Reaction (RT-PCR) testing. Recent studies show that community wastewater sampling can identify changes in COVID-19 incidence as low as 3.8 new cases per 100,000 [13], suggesting this limit is likely very low. These limitations mean that the probability of identifying any one infected individual on a single flight is very low [11]. As such, wastewater monitoring is not suitable for estimating the on-board prevalence of a disease. Rather, aircraft wastewater monitoring as a binary indicator of the presence of a pathogen on board a flight is more appropriate.

Unlike with respiratory swabbing, aircraft wastewater monitoring is not suitable for the identification, containment and contact tracing of individual infections on board a flight. Rather, it should be viewed as a means of providing public health authorities with information about the cross-border movement of pathogens. One such use for this might be to estimate the point-in-time of incursion of a new pathogen (or variant of a pathogen) into a country or region—potentially providing key insights for disease modelling and further health protection measures. To assess the feasibility of using aircraft wastewater in such a capacity the likely delay between the first incursion of a pathogen and its detection in wastewater must be first understood. A further application may be to facilitate a better understanding of global dynamics of disease that would be valuable for national health protection agencies–this is particularly useful to inform on transmission risk from low- and middle-income countries where disease surveillance may be limited.

Aircraft wastewater monitoring can have lower implementation costs than clinical testing regimes. While the per-sample cost of wastewater-based epidemiology can be higher than for clinical testing, as samples have to undergo several concentration steps, the overall cost can be reduced as a single sample can test for the presence of a pathogen in a pool of hundreds of individuals [14]. As wastewater monitoring is non-invasive, the impact on passengers is limited and the requirements for trained medical staff are reduced in comparison to individual testing (e.g. nasopharyngeal swabbing).

While air travel is an important factor in cross-border transmission of disease [1], it is important to note that other routes exist. This is especially pertinent in large multi-state or multi-nation land masses with good transport networks, like Europe and North America, where international

travel via land and waterways are common. Research into the feasibility of wastewater monitoring on cruise ships has yielded positive results [15], although this may not be directly comparable to air travel. Further work is needed to understand the functionality of wastewater based epidemiology across land borders, for example through monitoring of trains and transit hubs. Nevertheless, air travel remains a significant mode of international travel in much of the world and is responsible for a sizable proportion of cross-border disease transmission.

This paper aims to establish the likely length of any delay between incursion and detection, estimate the cumulative cases in the destination country at the point of detection, and to compare the effectiveness of aircraft wastewater monitoring to post-arrival respiratory swab screening. We compare wastewater monitoring to a random sample respiratory swab screening approach in which selection for screening is independent of infection state. This is similar to many border screening programmes implemented through 2022, as stringent pre-departure/post-arrival screening regimes were halted, such as the voluntary testing of travellers arriving from China into in the US, UK, Japan and Australia in late 2022 and early 2023 [16–18], We have considered SARS-CoV-2 as the target pathogen for our analysis as it has previously been the target of a large number of airport wastewater monitoring studies [8–10,15]; however, the results may be applicable to a wide range of diseases.

We use a two-patch infectious disease model to investigate the possible outcomes of aircraft wastewater monitoring. The model simulates flights between a seed (origin) country and the UK and uses a probabilistic approach to identify the first point at which wastewater monitoring identifies the presence of infected individual(s) on board. We then compare these results to simulated respiratory swab screening, in which a proportion of the passengers are tested directly for SARS-CoV-2 at the border.

Our results shed light on some of the practical considerations when using aircraft wastewater monitoring for public health surveillance. By comparing to respiratory swab screening regimes, we are able to highlight how wastewater monitoring can act as a cost effective, non-invasive approach to border pathogen surveillance with minimal impact on airline passengers.

## 2. Methods

### 2.1. SEI(F)R model

We use an SEIR (Susceptible, Exposed, Infectious, or Recovered) model to simulate an unmitigated epidemic in a seed country, as detailed in Eqs 1–5. The model is parameterised to have similar dynamics to SARS-CoV-2, with a latent period of $\alpha^{-1} = 5.2$ days, a recovery period of $\gamma^{-1} = 8$ days and $R_0 = \gamma/\beta = 2.5$ [19], where $\beta$ is the transmissibility constant. Importantly, the infected compartment is split into two stages; an initial infected state, denoted $I$, in which individuals can transmit the disease and test positive in both respiratory swabs and faecal samples, and a faecal positive state, denoted $F$, in which individuals are no longer infectious and test negative in respiratory swabbing but continue to test positive in faecal samples.

$$S'(t) = -\beta SI/N \tag{1}$$

$$E'(t) = \beta SI/N - \alpha E \tag{2}$$

$$I'(t) = \alpha E - \gamma I \tag{3}$$

$$F'(t) = \gamma I - \delta F \tag{4}$$

$$R'(t) = \gamma F \tag{5}$$

While there is evidence that a post symptomatic faecal shedding state exists for SARS-CoV-2 [1–3], there is little consensus on the mean length of time people spend in this state, denoted $\delta^{-1}$ in the model, with estimates of between 3 and 35 days provided in the literature [1,3]. As such, we consider how the model evolves over a range of values for $\delta$.

## 2.2. Air-travel model

We simulate two epidemics, one in a generic seed country with population $N_1 = S_1+E_1+I_1+F_1+R_1 = 10^6$, seeded with a single infection at $t = 0$, and an initially unseeded epidemic in the UK with population $N_2 = S_2+E_2+I_2+F_2+R_2 = 6.7\times10^7$. Each day, we simulate a single return trip between the two countries by taking a sample of $n$ people from each population and transferring them to the other population. By doing this we assume that aircraft passengers are a representative sample of the current state of the epidemic in their country of origin, and that no pre-departure screening of passengers occurs.

## 2.3. Detection of viral RNA in aircraft wastewater

Each flight contains some number of people in each disease state, with the total number of people on the flight given as $n = n_S+n_E+n_I+n_F+n_R$. As there is no pre-departure screening within our model, the probability of a single passenger being in each state is equal to the prevalence of that state in the country of origin. Each passenger has a probability $p_d$ of defecating during the flight, and defecating passengers in the $I$ and $F$ compartments have a probability $p_s$ of faecal shedding. Hence, the total number of stools produced during the flight is drawn from the binomial distribution $s\sim B(n, p_d)$, of which $s_{IF}\sim B(s,(n_I+n_F)/n)$ come from individuals in states $I$ and $F$, and $s_p\sim B(s_{IF}, p_s)$ test positive for SARS-CoV-2. The model assumes that the probability of defecation is independent of an individuals' infection state.

## 2.4. Limit of detection

There is a theoretical concentration of target RNA below which wastewater testing is unable to reliably identify the presence of the target pathogen. Changes in test-confirmed COVID-19 case rates as low as 3.8 new cases per 100,000 population can be identified in community wastewater with 99% probability [13]. However, this value is likely an under estimation of the true LoD as it relies only on test positive cases (admittedly over a period with very good test coverage), and is a measure of changes in incidence, rather than of prevalence. Further, it should be noted that aircraft wastewater is much more highly concentrated and specific than community wastewater, so detection is likely easier [8].

The lowest possible (non- zero) concentration of viral RNA achievable in our model occurs when every passenger on board deposits a stool into the wastewater, with just one of those stools containing viral RNA. In this 'worst-case' scenario, the LoD would have to be several orders of magnitude lower than that found previously [13] for sampling to return a negative result. This is corroborated in a recent study in which aircraft wastewater samples were compared against PCR screening of passengers, finding that positive wastewater samples were returned when a single infected individual was on board [8].

As such, we initially assume that wastewater sampling is sensitive enough to return a positive result when a single infected individual successfully transfers viral RNA into an aircrafts' wastewater. We also consider the sensitivity of our results to limits of detection of $0.0025 \leq L \leq 0.03$, measured as the ratio of infected stools, $s_p$, to total stools, $s$, in the wastewater system. In such cases, wastewater samples only test positive if $s_p/s \geq L$.

## 2.5. Respiratory swab screening

To assess the performance of aircraft wastewater monitoring we compare our results to the theoretical performance of post-arrival screening via respiratory swabbing of passengers. Respiratory testing is limited to capturing individuals in the $I$ compartment and does not detect those in the $F$ compartment. We consider scenarios in which the proportion of passengers screened is $0.1 \leq \phi \leq 0.5$, with test sensitivity of $0.65 \leq \eta \leq 0.99$. We draw the number of infected individuals selected for respiratory swabbing from a binomial distribution such that $r_I \sim B(n_I, \phi)$ infectious passengers are tested, resulting in $r_p \sim B(r_I, \eta)$ positives. We do not consider the occurrence of false positives in either the respiratory swabbing or in wastewater testing.

## 2.6. Initial parameterisation

For the initial results, we simulate the model 10,000 times using the parameterisation described in Table 1. Model parameters $\alpha$, $\beta$, $\gamma$, $L$, $\phi$ and $\eta$ are kept constant through all simulations. We draw the values of $p_d$ and $n$ from uniform distributions such that $0.1 \leq p_d \leq 0.6$ and $100 \leq n \leq 400$, reflecting flights of different lengths and different passenger numbers. Due to the lack of consensus in the literature about the length of the post symptomatic faecal shedding period, $\delta^{-1}$, this value is also drawn from a uniform distribution of integer values such that $3 \leq \delta^{-1} \leq 28$. We draw the faecal shedding probability, $p_s$, from a beta distribution such that $p_s \sim Beta(a, b) \in [0,1]$, where:

$$a = \left( \frac{1 - \mu}{\sigma^2} - \frac{1}{\mu} \right) \mu^2,$$

and

$$b = a \left( \frac{1}{\mu} - 1 \right),$$

where $\mu = 0.459$ and $\sigma = 0.168$ are the mean and standard deviation of the distribution [2].

**Table 1. Model parameterisation for results in Fig 1.** The post symptomatic faecal shedding period, $\delta^{-1}$, number of people on flight, n and probability of defecation, $p_d$, are drawn from uniform distributions. The faecal shedding probability, $p_s$, is drawn from a Beta distribution with parameters a and b set such that the distribution has mean 0.459 and standard deviation 0.168.

| Parameter | Description | Value |
|---|---|---|
| $\alpha$ | Latency rate | $5.2^{-1}\ days^{-1}$ |
| $R_0$ | Reproductive number $(\beta/\gamma)$ | 3.0 |
| $\gamma$ | Recovery rate | $8.0^{-1}\ days^{-1}$ |
| $\delta$ | Post symptomatic faecal shedding recovery rate | $\delta^{-1} \sim U(3,28) days$ |
| $n$ | Number of people on flight | $n \sim U(100,400)$ |
| $p_d$ | Probability of defecation | $p_d \sim U(0.1,0.6)$ |
| $p_s$ | Probability of faecal shedding | $p_s \sim Beta(a,b)$ |
| $L$ | Limit of detection (LoD) | 0.005 |
| $\phi$ | Proportion screened using respiratory swabs | 0.20 |
| $\eta$ | Sensitivity of respiratory swabs | 0.85 |

## 2.7. Parameterisation for sensitivity analysis

When investigating the sensitivity of the model to specific parameters all other parameters are held constant. For these analyses we set $n = 250$, $\delta^{-1} = 10$ days, $p_d = 0.36$, $p_s = 0.459$ (as used in [11]), and $\eta = 0.3$; all other parameters are as described in Table 1. This parameterisation minimises the delay between the median date of detection in wastewater and respiratory swab screening to zero days.

## 3. Results

### 3.1. Monte-Carlo simulation with broad distribution of parameters

Fig 1 shows the time of first seeding into the UK, the time of first detection in aircraft wastewater, and the time of first detection in respiratory swabs from 10,000 simulations of the model with the parameterisation described in Table 1. The analysis suggests that, for the given range of parameters, the median date of detection in wastewater is two days earlier than for respiratory swabs (wastewater median 94 days, inter-quartile (IQ) range 85–102 days; respiratory swab median 96 days, IQ range 88–103 days). There was a median delay of 17 days (IQ range: 7–28 days) between first seeding in the UK and detection in wastewater, resulting in a median of 25.2 (IQ range: 4.8–78.2) cumulative infections in the UK, including imported cases. At the point of detection via respiratory swabbing there were a median of 33.4 (IQ range: 9.3–24.2) cumulative infections in the UK.

### 3.2. Effect of screening coverage and sensitivity

Fig 2 shows the delay between detection in wastewater and detection using respiratory swabs across screening coverages $\varphi \in \{0.1, 0.2, 0.3, 0.4, 0.5\}$ and screening sensitivities $\eta \in \{0.65, 0.75,$

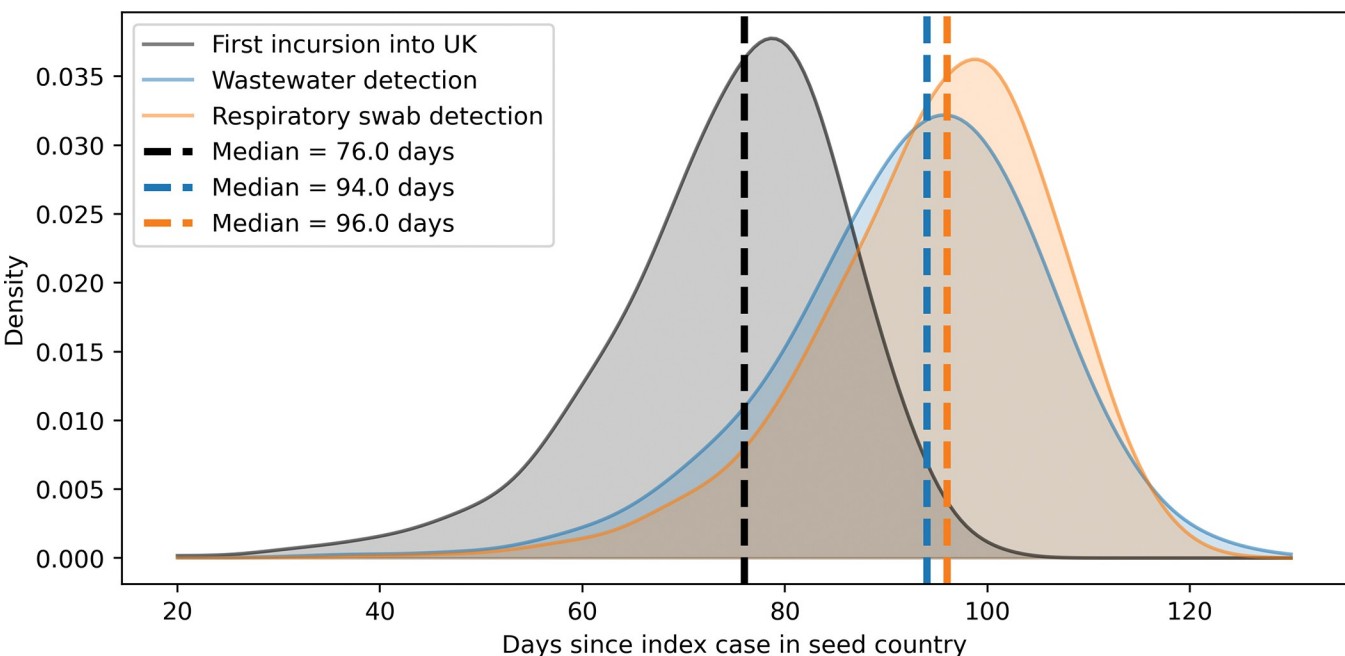

**Fig 1. The distribution of results from 10,000 simulations of the model, using the parameter ranges given in Table 1.** The black distribution is the time of first incursion in the UK, given as the number of days since the index case in the country of origin. The blue and orange distributions show the time of detection in wastewater and respiratory swabbing, respectively. The dashed lines show the median value for each distribution.

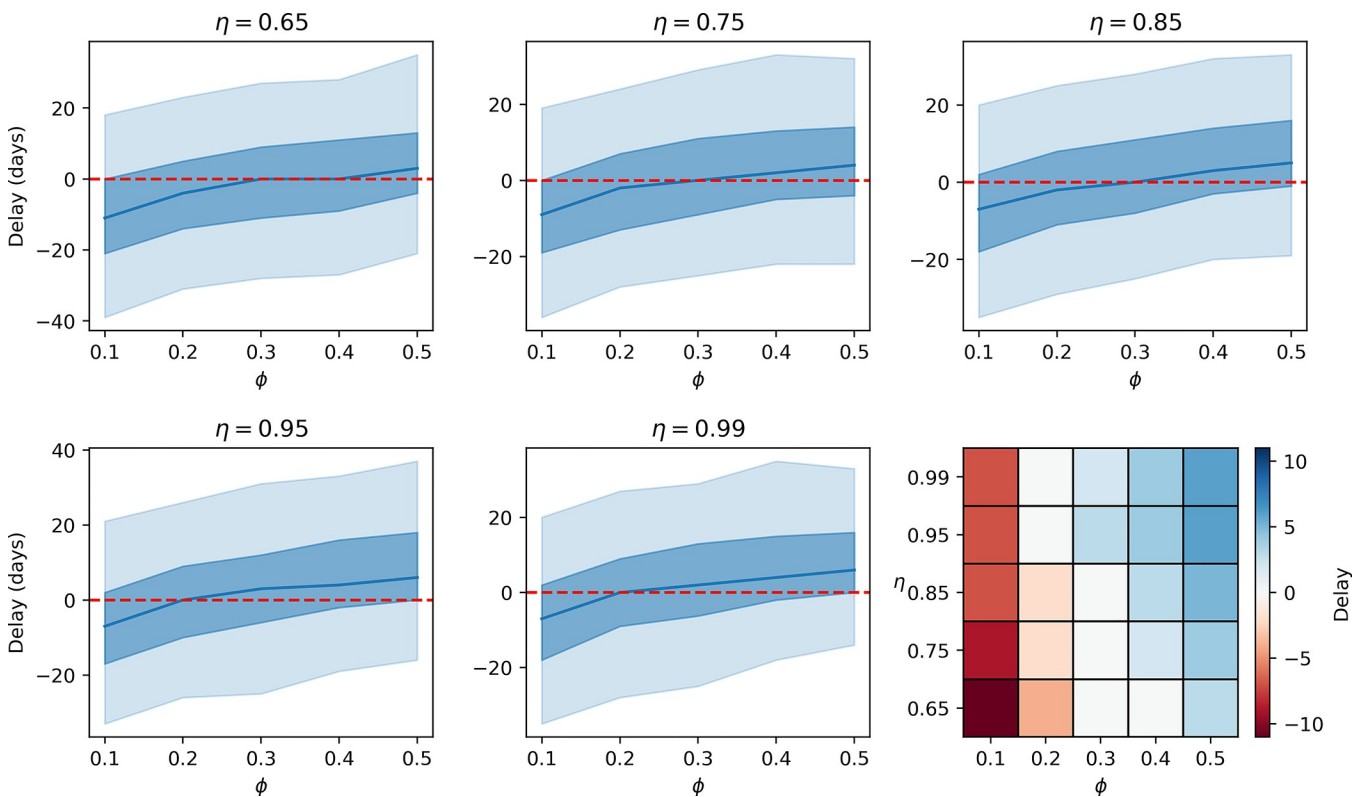

**Fig 2. Delay between detection in wastewater and detection via respiratory swabs for different values of respiratory swab coverage (ϕ) and sensitivity (η) over 2000 model simulations.** Other variables are as indicated in Table 1. A negative value indicates detection in wastewater occurs prior to detection in respiratory swabs. The central blue line gives the median value of 2000 simulations of the model, the blue shaded areas give the 25–75% and 5–95% intervals. A comparison of the median delay across each value of η and ϕ is shown in the heat–map in the final plot.

0.85, 0.95, 0.99} over 2000 model simulations. All other parameter values and ranges are as indicated in Table 1. A negative value indicates that wastewater detection occurs before detection via respiratory swabs. For test sensitivities $\eta \leq 0.85$, a screening coverage of $\phi \geq 0.3$ is required to outperform wastewater testing in more than half of the simulations. As test sensitivity increases (for example by using nasopharyngeal swabbing rather than nasal swabbing) this proportion falls to $\phi \geq 0.2$ for $\eta > 0.95$.

### 3.3. Sensitivity to model parameters

Fig 3 shows the delay between detection in wastewater and in respiratory swab screening across a range of values for LoD, defecation probability, shedding probability and post symptomatic faecal shedding period for 2000 model simulations. Other parameter values remain the same as stated in section 2.7.

Analysis of the LoD parameter, $L$, suggests that values of $L \leq 0.01$ have little effect on model output. However, as LoD increases, a significant reduction in performance relative to respiratory swabbing is observed. Nevertheless, as discussed previously it is likely that the true LoD is extremely low—with the estimate by Li *et al.* suggesting $L < 0.005$ [13].

Defecation probabilities of $p_d < 0.3$ result in detection in respiratory swabbing earlier than in wastewater in more than half of simulations. For $p_d \leq 0.1$ respiratory swabs outperform wastewater monitoring in around 75% of simulations. As $p_d$ is directly correlated with flight

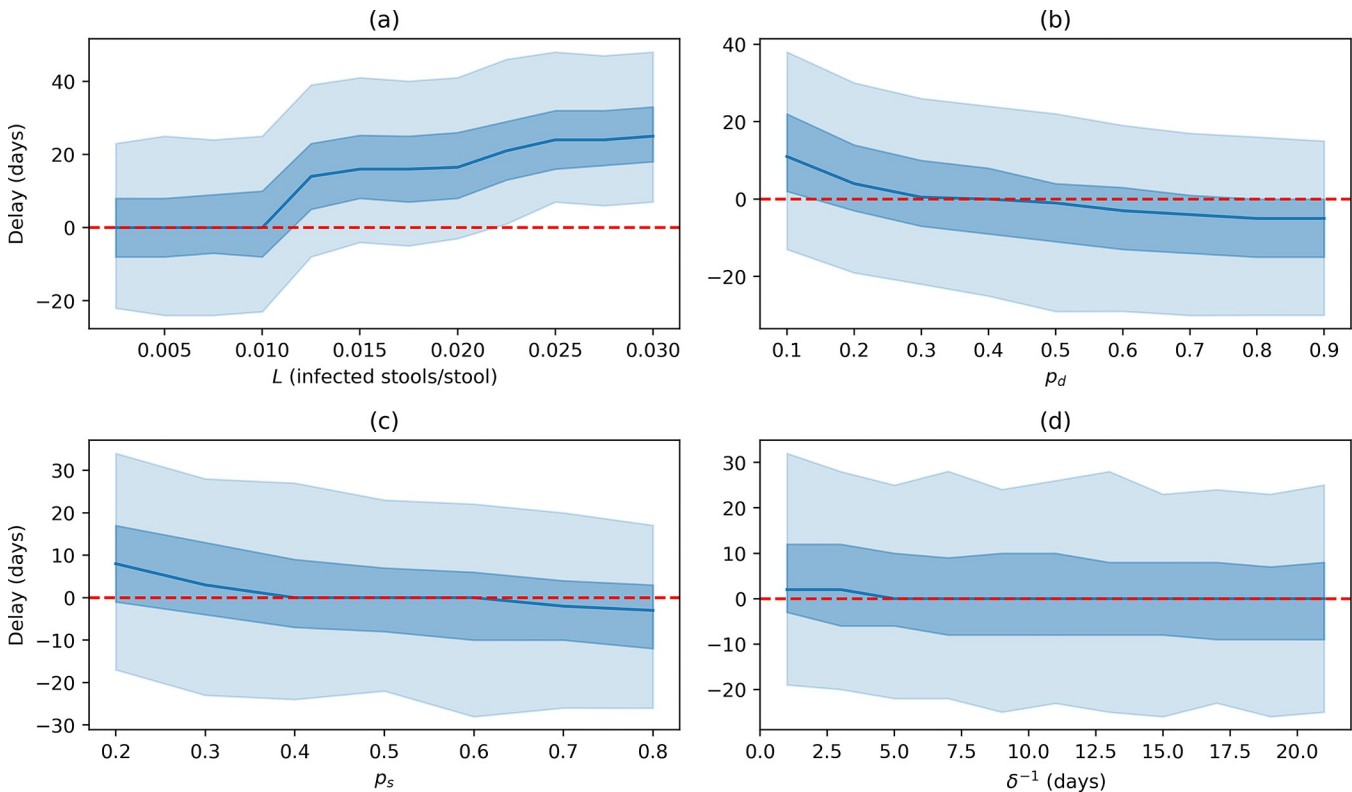

**Fig 3.** Model outputs for 1000 model simulations across a range of values of (a) LoD, (b) defecation probability, (c) shedding probability and (d) post–symptomatic faecal shedding period. All other parameters are as stated in section 2.6. The central blue line shows the median delay between detection in wastewater and in respiratory screening, the blue shaded areas represent the 25–75% and 5–95% intervals.

duration we can infer that wastewater monitoring is likely to be more effective for longer flights.

The faecal shedding probability, $p_s$, and post-symptomatic faecal shedding period, $\delta$, relate to the characteristics of the disease. Analysis of these parameters may give an indication of the effectiveness of wastewater monitoring in the surveillance of other diseases. Wastewater monitoring is less effective than respiratory swab screening when $p_s < 0.4$ and slightly more effective for $p_s > 0.6$. Changes in the post-symptomatic faecal shedding period have very limited influence on the effectiveness of wastewater monitoring. This is likely because the total number of people in the $I$ and $F$ compartments at the point of detection is very low so the influence of $\delta$ is minimal. It is likely that later in the epidemic $\delta$ would have a more significant effect on detection.

## 4. Discussion

As support for strict border control measures reduces, public health authorities are left with limited choices in how to monitor the incursion of SARS-CoV-2, and other infectious diseases, across borders. This paper has investigated the extent to which aircraft wastewater monitoring might be able to bridge the gap left by reduced respiratory screening. We have shown that, for a broad range of parameterisations of our model, random sample respiratory swab screening begins to outperform wastewater monitoring when screening covers around 20% of passengers. However, sensitivity analysis has identified several factors which should be considered when using wastewater monitoring as a surveillance tool.

The length of flight is a clear influence on the viability of aircraft wastewater monitoring since individuals are more likely to defecate during a longer flight [11]. Unfortunately, there is no reliable data on the specific relationship between flight time and probability of defecation, although one can reasonably assume that the relationship is positive. Further work should aim to estimate the value of $p_d$ for a given flight-time, based on the distribution of times of day individuals tend to defecate [20].

The sensitivity of the wastewater sampling technique is also important. For sampling methods which increase the limit of detection above 0.01 infected stools per total stools we see a sharp reduction in the performance of wastewater monitoring. Further work should attempt to identify this value in a biologically relevant unit, such as gene copies per ml, however this is outside the scope of this work.

The disease, or variant of disease, may also play a role in the practical application of aircraft wastewater monitoring. The probability of shedding viral RNA into faeces has a significant effect on the viability of wastewater monitoring—a disease with faecal shedding probability of $p_s < 0.4$ is unlikely to be suitable for wastewater monitoring. The sensitivity analysis was unable to identify a strong relationship between post-symptomatic faecal shedding period and performance of wastewater monitoring for the parameter ranges considered in the analysis.

There is typically some delay between the sampling of aircraft wastewater and the sample being tested through RT-qPCR. We have not considered this delay in our model, instead assuming a test can be returned instantaneously. For cases where aircraft wastewater is being used to retrospectively estimate the point in time of incursion of a disease this assumption is valid, however the delay should be considered for cases where wastewater is being used in a more reactive capacity. Respiratory screening at the border can return (almost) instant indications of traveller positivity—although in practice during the SARS-CoV-2 pandemic many countries opted for testing recently arrived passengers a set number of days after arrival.

This paper has only considered the time to detection as a metric for measuring the value of wastewater monitoring. While we have shown that wastewater can be more effective than limited respiratory screening at detecting the presence of SARS-CoV-2 on board flights, there are other factors which have not been considered. One of the strengths of wastewater monitoring is that sample taking is agnostic to the target pathogen, so a single sample can be tested for multiple diseases. As the same degree of flexibility is not afforded by respiratory swabbing it may be that wastewater monitoring is preferred to border screening even in situations where screening would likely detect the presence of a pathogen more quickly.

As we move away from strict border controls as a means of mitigating the international transmission of pathogens like SARS-CoV-2 it is important that we understand the tools which remain available to us. The results discussed in this paper lay some of the theoretical groundwork in doing that for aircraft wastewater monitoring. We have shown that, under specific circumstances, aircraft wastewater monitoring can be an effective tool in global public health surveillance.

## 5. Conclusion

The model presented in this paper provides valuable insights into the potential viability of aircraft wastewater monitoring as a tool for public health surveillance. We have shown that, for simulations across a broad parameterisation of our model, there is a delay of around 17 days between the first incursion of a SARS-CoV-2 like pathogen and its detection in aircraft wastewater, and that this delay results in a median of 22 cumulative infections in the UK. Further, we have shown that wastewater screening can detect a new pathogen earlier than respiratory swab screening of 30% of passengers using a test with 85% sensitivity. This result, however, is

dependent on a number of factors, with the probability of defecation during a flight (assumed to be correlated to flight time), the probability of shedding pathogen into faeces and the sensitivity of wastewater testing all having significant influence on the viability of wastewater monitoring.

## Acknowledgments

We would like to thank the Advanced Analytics Division of the Data, Analytics and Surveillance Department at UKHSA for their helpful comments and suggestions during development of this model, and Elena Karakashevska for her help with assessing the cost effectiveness of aircraft wastewater monitoring.

## Author Contributions

**Conceptualization:** Joseph W. Shingleton, Matthew J. Wade.

**Formal analysis:** Joseph W. Shingleton.

**Investigation:** Joseph W. Shingleton.

**Methodology:** Joseph W. Shingleton.

**Project administration:** Joseph W. Shingleton.

**Software:** Joseph W. Shingleton.

**Supervision:** Chris J. Lilley, Matthew J. Wade.

**Validation:** Joseph W. Shingleton.

**Visualization:** Joseph W. Shingleton.

**Writing – original draft:** Joseph W. Shingleton.

**Writing – review & editing:** Joseph W. Shingleton, Chris J. Lilley, Matthew J. Wade.

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
