## [Decision Letter · Decision Letter 0]

20 Apr 2023

PGPH-D-23-00441

Evaluating the theoretical performance of aircraft wastewater monitoring as a tool for SARS-CoV-2 surveillance

Dear Dr. Joseph Singleton,

Thank you for submitting your manuscript to PLOS Global Public Health. After careful consideration, we feel that it has merit but does not fully meet PLOS Global Public Health’s publication criteria as it currently stands. Therefore, we invite you to submit a revised version of the manuscript that addresses the points raised during the review process.

We look forward to receiving your revised manuscript.

Kind regards,

Muhammad Asaduzzaman, MD MPH MPhil

Academic Editor

Journal Requirements:

Additional Editor Comments (if provided):

Reviewers' comments:

Reviewer's Responses to Questions

**Comments to the Author**

1. Does this manuscript meet PLOS Global Public Health’s publication criteria? Is the manuscript technically sound, and do the data support the conclusions? The manuscript must describe methodologically and ethically rigorous research with conclusions that are appropriately drawn based on the data presented.

Reviewer #1: Yes

Reviewer #2: Yes

Reviewer #3: Yes

2. Has the statistical analysis been performed appropriately and rigorously?

Reviewer #1: I don't know

Reviewer #2: I don't know

Reviewer #3: Yes

3. Have the authors made all data underlying the findings in their manuscript fully available (please refer to the Data Availability Statement at the start of the manuscript PDF file)?

Reviewer #1: Yes

Reviewer #2: No

Reviewer #3: Yes

4. Is the manuscript presented in an intelligible fashion and written in standard English?

Reviewer #1: Yes

Reviewer #2: Yes

Reviewer #3: Yes

5. Review Comments to the Author

Reviewer #1: GENERAL COMMENTS:

This innovative paper presents a computer simulation model for assessing the relative role, strengths, and weaknesses of aircraft wastewater monitoring versus airline passenger respiratory screening for surveillance of the SARS-CoV-2 pandemic. The article also discusses the possible broader use of aircraft wastewater monitoring for other infectious diseases. I have suggested brief additions to the Abstract, corrected spelling errors, and requested clarifications in sections of the paper. I am not an expert on simulation computer modeling, so I only provided limited comments on the computer model methods section.

SPECIFIC COMMENTS USING LINE NUMBERING IN THE MANUSCRIPT:

ABSTRACT:

-Line 10, please replace "import" with "important."

-In the final section of the ABSTRACT, I would suggest adding a sentence or two that states that in the Discussion section of the paper the authors provide insights on the suitability of aircraft wastewater monitoring based on a number of parameters, including sample availability, sample collection, test sensitivity and the characteristics of the target pathogen.(Lines 269 to 320.) This would allow the reader to assess the described computer modeling exercise in the broader context of infection disease surveillance.

INTRODUCTION:

In the opening section, the authors focus on the role of aircraft travel in disease dissemination. It may be useful to include a sentence or two on other cross border settings, such as in mainland Europe, where a preponderance of cross border travel may not be by air (for example, train travel)

Line 70, I would suggest deleting the word "case" in the sentence,"....One such use case......."

Line 93, please replace the word, "effect" with "effective."

DISCUSSION:

Line 269, I would suggest replacing the word, "appetite" with "support."

Reviewer #2: As an introduction to my review, I would like to point out that my evaluation concerns mainly the approach to prevention, surveillance and public health proposed by your paper (rather than the modelling work on the methodological level).

In terms of method:

- You compare the analysis of wastewater with the swabs taken on arrival. I did not understand in your text the status of the passengers who would be tested by swab: random selection? Symptomatic passengers? Passengers who would report contact with a case? Because the probability of being infected, and therefore the probability of being positive, varies significantly in these three groups. In practice, many territories carried out either a systematic screening before departure, or a screening after arrival of symptomatic persons. Random screening is not, to my knowledge, a common practice.

- You mention the fact that wastewater testing is cost effective compared to screening on arrival (under certain conditions of proportion of screened passengers and sensitivity of the test). I did not find any mention of cost analysis of the two measures in your methods section. Although intuitive, this statement does not seem to be supported by your results. Even if the costs of the analyses may vary from country to country, I think you could introduce in your work estimates made on the basis of the cost in UK, complementing in a simple way your results. (Or you could could remove the mention about cost-effectiveness)

In general, it seems to me that the most important issue is really not discussed in your article: what would be the objective, the benefit of such a screening? In the case of pre-departure screening, the objective was to prevent the travel of infectious subjects, reducing the probability of transmission during the flight and in the territory of arrival; in the case of arrival screening (systematic or of symptomatic or at-risk individuals) the objective was the quarantine of positive travellers (and contact tracing of on-board neighbouring persons), with the objective of reducing transmission in the country of arrival. These methods have significant costs (in financial and human terms), and important limitations (in terms of effective case detection capacity), but meet clear prevention objectives. What would be the benefit of wastewater testing? I would argue that it is not possible to carry out targeted prevention measures on the basis of these results. So it would be a way to estimate the risk of introduction for "purely" epidemiological purposes? (In this case, Sars-CoV2 might be the wrong example, because it is no more an emerging pathogen.) What kind of actions could be taken after a positive result?

I strongly suggest that you expand the introduction and discussion by making more explicit the (public health) rationale for the use of aircraft wastewater monitoring, and the context in which you think it would be relevant to use it.

Reviewer #3: The grammar in the entire manuscript has to be improved greatly. In the methodology section, there is a need to include a description of the study, that is, the two countries used for the simulation and predictions. There is indiscriminate use of italics and future tenses in the method section.

6. PLOS authors have the option to publish the peer review history of their article (what does this mean?). If published, this will include your full peer review and any attached files.

**Do you want your identity to be public for this peer review?** For information about this choice, including consent withdrawal, please see our Privacy Policy.

Reviewer #1: **Yes: **Paul R De Lay, MD, DTM&H (Lond)

Reviewer #2: No

Reviewer #3: No

---

## [Decision Letter · Decision Letter 1]

2 Jun 2023

Evaluating the theoretical performance of aircraft wastewater monitoring as a tool for SARS-CoV-2 surveillance

PGPH-D-23-00441R1

Dear Joseph Shingleton,

We are pleased to inform you that your manuscript 'Evaluating the theoretical performance of aircraft wastewater monitoring as a tool for SARS-CoV-2 surveillance' has been provisionally accepted for publication in PLOS Global Public Health.

Best regards,

Muhammad Asaduzzaman, MD MPH MPhil

Academic Editor

Reviewer Comments (if any, and for reference):

Reviewer's Responses to Questions

**Comments to the Author**

1. If the authors have adequately addressed your comments raised in a previous round of review and you feel that this manuscript is now acceptable for publication, you may indicate that here to bypass the “Comments to the Author” section, enter your conflict of interest statement in the “Confidential to Editor” section, and submit your "Accept" recommendation.

Reviewer #1: All comments have been addressed

Reviewer #2: All comments have been addressed

Reviewer #3: All comments have been addressed

2. Does this manuscript meet PLOS Global Public Health’s publication criteria? Is the manuscript technically sound, and do the data support the conclusions? The manuscript must describe methodologically and ethically rigorous research with conclusions that are appropriately drawn based on the data presented.

Reviewer #1: Yes

Reviewer #2: Yes

Reviewer #3: Yes

3. Has the statistical analysis been performed appropriately and rigorously?

Reviewer #1: Yes

Reviewer #2: I don't know

Reviewer #3: Yes

4. Have the authors made all data underlying the findings in their manuscript fully available (please refer to the Data Availability Statement at the start of the manuscript PDF file)?

Reviewer #1: Yes

Reviewer #2: Yes

Reviewer #3: Yes

5. Is the manuscript presented in an intelligible fashion and written in standard English?

Reviewer #1: Yes

Reviewer #2: No

Reviewer #3: Yes

6. Review Comments to the Author

Reviewer #1: I have reviewed the revised version of the paper, which appears as the second version in the attached document. Thanks to the authors for incorporating the suggested revisions. I recommend that the paper be accepted and published.

Reviewer #2: The authors deeply modified introduction and discussion.

Reviewer #3: Not applicable.

7. PLOS authors have the option to publish the peer review history of their article (what does this mean?). If published, this will include your full peer review and any attached files.

**Do you want your identity to be public for this peer review?** For information about this choice, including consent withdrawal, please see our Privacy Policy.

Reviewer #1: **Yes: **Paul R De Lay, MD, DTM&H (Lond)

Reviewer #2: No

Reviewer #3: **Yes: **Dr Uwem Edet
